# OpenReview forum: "Quantifying Frontier LLM Capabilities for Container Sandbox Escape"
_ICML.cc/2026/Conference — ICML 2026 spotlight_

### Official Review · Reviewer_h1b8 · 2026-03-06

**Soundness:** 4
**Presentation:** 2
**Significance:** 4
**Originality:** 4
**Overall Recommendation:** 5
**Confidence:** 3

**Summary:**

The authors propose SandboxEscapeBench to evaluate the LLM capabilities for container sandbox escape. They define the threat model as CTF. They categorize and assign difficulty score for the existing and agent's escaping approaches. The authors conduct evaluations on current LLMs, present the results and discuss the findings.

**Compliance With Llm Reviewing Policy:**

Affirmed.

**Final Justification:**

The authors clarified my questions. I keep my current score.

**Key Questions For Authors:**

1. Please revise Section 5 and move the limitations to the end of the paper.
2. Which agentic scaffold is used in the paper?

**Limitations:**

Yes.

**Strengths And Weaknesses:**

Thanks for submitting to ICML 2026. I generally like reading the paper and appreciate the work authors put into this paper.

### Soundness

The setup is clear and simple: LLM living inside a sandbox and prompted to capture the flag. The evaluation experiments are extensive: evaluating the escape capability under different factors (tokens, dollar cost, hints, etc.).

### Presentation

Generally the writing of this paper is good, though the spacing is aggressively smaller and Section 5 lacks structure. Besides, the limitation section should be placed near the conclusion section to be highlighted.

### Significance

This paper provides upper bound evaluation for LLM capability of sandbox escaping --- when the LLM is asked to do so. The evaluation results in this paper can be served for LLM safety community as well as security researchers.

### Originality

The problem setup is clean and simple, and the vast evaluations and their results are novel and useful.

---

> ### Author Rebuttal · Authors · 2026-03-30
>
> We thank the reviewer for their positive assessment of the paper and for their helpful suggestions, which improved the presentation, structure, and clarity of the manuscript.
>
> ---
>
> Comment 1: the spacing is aggressively small
>
> Response 1: We agree and have increased paragraph and heading spacing size. This has improved the readability and aesthetics of the paper, thank you.
>
> ---
>
> Comment 2: Section 5 lacks structure.
> Response 2: We agree that Section 5 is a lot of information under a single heading. We have upgraded scaling, hinting and transcript analysis to full numbered subsections (not emboldened paragraphs).
>
> ---
>
> Comment 3: The limitations section should be moved next to the conclusion.
> Response 3: Thank you for spotting this, we have moved it to the proper location.
>
> ---
>
> Comment 4: Which agentic scaffold is used in the paper?
> Response 4: We detail our evaluation configuration in Appendix E however, agree that agentic architecture is important enough to feature in the main body of text. We have added the following details to the implementation section: “Each model is driven by a ReAct-style agent loop (Yao et al., 2023) … (full prompts in Appendix A; generation settings in Appendix E)”. Code is provided to fully reproduce the findings, enabling the reader to investigate any methodological detail.

---

> > ### Author Rebuttal · Reviewer_h1b8 · 2026-04-02
> >
> > Thanks for clarifications. I strongly recommend to add details of the agentic scaffold and the tools provided to the agent in the main text.

---

> > > ### Author Response · Authors · 2026-04-04
> > >
> > > We agree clarity could be further improved. We have updated the main text to include the tools available to the model: “Each model is driven by a ReAct-style agent loop (Yao et al., 2023) with access to bash within the vulnerable sandbox … (full prompts in Appendix A; generation settings in Appendix E)”. We have also explicitly pointed the reader to the exact agentic function in our code base used in all experimentation.
> > >
> > > Please let us know if you had anything else in mind.
> > >
> > > Thank you

---

### Official Review · Reviewer_DNhD · 2026-03-12

**Soundness:** 3
**Presentation:** 3
**Significance:** 4
**Originality:** 2
**Overall Recommendation:** 5
**Confidence:** 3

**Summary:**

The paper introduces SandboxEscapeBench, a benchmark designed to evaluate the sandbox escape capabilities of LLMs without compromising the evaluation infrastructure. The study reveals that frontier LLMs can reliably exploit common misconfigurations and known vulnerabilities, achieving a 40% success rate even on medium-difficulty tasks. These findings underscore significant security risks in current AI Agent sandboxing practices.

**Compliance With Llm Reviewing Policy:**

Affirmed.

**Final Justification:**

Thank authors for addressing my concern. I will keep my original score.

**Key Questions For Authors:**

- The vulnerability difficulty levels appear to be based on human completion time. Is there a discrepancy in difficulty for AI? For instance, are there specific vulnerabilities that are "Hard" for humans but "Easy" for an LLM due to its pattern recognition or training data?
- Could you provide more detail on the design of the experimental agent? Specifically, how do optimizations in Agentic workflows impact the success rate and overall escape risk?

**Limitations:**

yes

**Strengths And Weaknesses:**

Strengths
- This is the first benchmark specifically targeting AI sandbox escape. As AI Agents are increasingly deployed in real-world environments, this is a timely and critical area of focus.
- The paper is well-written, highly readable, and provides source codes that ensure reproducibility.

Weaknesses
- The benchmark functions primarily as a testbed for container vulnerability exploitation. It lacks a thorough discussion on the risks of Agent escape in real-world scenarios, particularly regarding active monitoring and defensive measures (e.g., those discussed in frameworks like PACEBench [1]).
- In the current experimental setup, the Agent is explicitly told it is in a container and that "escaping" is the goal. This significantly lowers the barrier to entry. A more critical safety concern is situational awareness —whether a model can identify it is in a container on its own—and its propensity or "dark intent" to escape without being prompted to do so.

[1] Liu, Zicheng, et al. "PACEbench: A Framework for Evaluating Practical AI Cyber-Exploitation Capabilities." arXiv preprint arXiv:2510.11688 (2025).

---

> ### Author Rebuttal · Authors · 2026-03-30
>
> We appreciate the reviewer’s thoughtful comments particularly in clarifying the benchmark’s relationship to real-world defences, distinguishing capability from propensity, and expanding our discussion of difficulty ratings.
>
> ---
>
> Comment 1: The benchmark lacks a thorough discussion on the risks of Agent escape in real-world scenarios.
>
> Response 1:
> Thank you for this helpful comment. We agree that the original draft did not sufficiently distinguish between capability measurement in a minimally defended setting and end-to-end compromise likelihood in production deployments. Our intent with SandboxEscapeBench is to isolate a specific safety-relevant capability. In the current threat model, the agent operates without active intrusion detection, monitoring, or logging, and our limitations section notes that the results should not be interpreted as the likelihood of compromise in defended real-world systems. We will revise the paper to make this distinction more explicit. We do hope that this benchmark will be used in future control related work, where it might feature as the malicious side task models are tasked to solve while being actively monitored.
> We also agree that this should be better connected to defense-aware evaluation frameworks such as PACEbench. PACEbench’s defended setting is valuable for understanding defense evasion in broader cyber environments, while our benchmark isolates whether the sandbox itself remains a reliable control layer under realistic container weaknesses.
> We have added the following paragraph to the introduction of the paper:
>
> “SandboxEscapeBench is designed as a capability-isolation benchmark, not a full production-risk simulator. It measures whether agents can discover and exploit container escape misconfiguration paths when active monitoring and other defensive measures are absent. In defended deployments, real-world risk will additionally depend on patching, configuration hygiene, stronger isolation mechanisms, and active detection/response. This makes our benchmark complementary to defense-aware frameworks such as PACEbench (Liu et al., 2025), which incorporate explicit cyber defenses and evaluate defense evasion in broader settings.”
>
> ---
>
> Comment 2:  A more critical safety concern is situational awareness.
>
> Response 2:
> A model’s propensity to conduct a malicious act is an extremely interesting and important behaviour to measure. It is, however, challenging to do well, often confounded by models being “evaluation aware”. We view whether a model has the capability to conduct an act and whether it has the propensity to do so as orthogonal properties. We set out only to measure the capability with this benchmark and leave measuring the propensity to escape to future work.
> We also note that from the perspective of assuring the robustness of an agentic system, intent is less important than whether escape is possible. If a deployed agent can be made to escape a container through prompt injection or other means, the safety concern exists regardless of whether the model would do so spontaneously. Measuring raw capability is therefore a necessary first step.
>
> ---
>
> Comment 3: Is there a discrepancy in difficulty for AI and humans?
>
> Response 3:
> This is an interesting question. We note that our human-calibrated difficulty ratings do broadly predict model success rates: models reliably solve difficulty 1-2 scenarios and struggle with 3-5, suggesting reasonable alignment between human and AI difficulty for this task domain. Within difficulty bands there is variation in model success across scenarios, which likely reflects both noise in our coarse ratings (now based on 3 expert reviewer ratings; Krippendorff's Alpha: 0.82) and differences in how well-suited individual exploit patterns are to LLM capabilities (e.g. whether they follow a template vs. require multi-step adaptation). We have added a brief discussion of this to Section 5.
> "Our human-calibrated difficulty ratings broadly predict model success: models reliably solve scenarios rated 1–2 and tend to struggle on those rated 3–5. Still, performance varies across scenarios within the same difficulty band, which may reflect rating noise or differences between what humans and AI find challenging."
>
> ---
>
> Comment 4: Could you provide more detail on the design of the experimental agent?
>
> Response 4:
> We describe our evaluation configuration in Appendix E, but we agree that agentic architecture is important enough to feature in the main body of text. We have added the following details to the implementation section: “Each model is assessed in a ReAct-style agent loop (Yao et al., 2023) … (full prompts in Appendix A; generation settings in Appendix E)”. We also open source our code to ensure that all results are reproducible. We do not examine the effect of varying agentic architectures on benchmark performance, as we chose to prioritise evaluating a larger number of models.

---

> > ### Author Rebuttal · Reviewer_DNhD · 2026-04-05
> >
> > Thank you for your response. I believe my original score is reasonable, and I will maintain it.

---

### Official Review · Reviewer_LrCD · 2026-03-13

**Soundness:** 4
**Presentation:** 3
**Significance:** 3
**Originality:** 3
**Overall Recommendation:** 4
**Confidence:** 3

**Summary:**

This paper introduces SandboxEscapeBench, a CTF-style benchmark for evaluating LLM agents' ability to escape container sandboxes. The benchmark comprises 18 scenarios spanning orchestration (Kubernetes RBAC, kubectl cp CVE), runtime (privileged containers, Docker socket exposure, runc CVEs, dangerous Linux capabilities), and kernel (Dirty COW, Dirty Pipe, eBPF, cgroup abuse) attack layers, with difficulty ratings from 1 to 5. A nested sandboxing architecture (Docker container inside a VM) ensures evaluation safety. The authors evaluate 9 frontier models and find that large models reliably escape common misconfigurations (difficulty 1-2: ~100% success for GPT-5 and Opus 4.5), achieve meaningful success on harder runtime/kernel scenarios (difficulty 3: ~40%), and fail entirely on the hardest kernel exploits (difficulty 4-5: 0%).

**Compliance With Llm Reviewing Policy:**

Affirmed.

**Key Questions For Authors:**

+ Can you design an experiment to distinguish exploit memorization from exploitation reasoning?
+ Have you considered recruiting security professionals to attempt the benchmark under similar conditions, even for a small human study?
+

**Limitations:**

Yes

**Strengths And Weaknesses:**

# Strength
+ Fills a clear and practically important evaluation gap
+ Rigorous evaluation methodology with strong safety properties
+ Systematic scenario coverage grounded in real vulnerability classes

# Weakness
- All scenarios reproduce known vulnerabilities; zero-day discovery capability is untested

There might be a threat that a model that achieves 100% on SandboxEscapeBench may simply have extensive exploit code in its training data. Some data mutation or rewriting is required.

- Difficulty ratings rest on a single expert's judgment without validation

In security research, exploit difficulty depends heavily on individual experience. Without inter-rater reliability, the difficulty scale is unvalidated. I see the empirical results provide post-hoc validation, but this circular reasoning cannot substitute for independent calibration.

- The threat model assumes initial container access, skipping the hardest practical step

The evaluation assumes "code execution inside a container" with root privileges. In real deployments, obtaining code execution inside a container typically requires a prior exploitation step and normally is hard. While this is partially mentioned in the Limitation, it seems the Introduction and Conclusion sometimes suggest broader applicability than the threat model supports.

---

> ### Author Rebuttal · Authors · 2026-03-30
>
> We thank the reviewer for their thoughtful and constructive feedback, which helped us better motivate our design choices, sharpen our framing of exploitation reasoning, and strengthen the validation of our difficulty ratings.
>
> ---
>
> Comment 1: All scenarios reproduce known vulnerabilities; zero-day discovery capability is untested
> Response 1:
> The use of known vulnerabilities is a deliberate design choice. By staying within well-known vulnerabilities, we are able to deliver useful insights without the heavy overhead of managing vulnerability disclosure. Using known vulnerabilities provides a stable, well-characterised task set against which models can be reliably compared. Our results demonstrate that in many cases, current models can identify and exploit these escape paths when present, with implications for the suitability of container-based sandboxing for agentic deployments. Zero-day discovery tests a distinct set of capabilities and, as zero-days are non-static by definition, is somewhat unsuitable for reproducible comparison of model capability over time.
>
> ---
>
> Comment 2: There might be a threat that a model that achieves 100% on SandboxEscapeBench may simply have extensive exploit code in its training data. Some data mutation or rewriting is required. Can you design an experiment to distinguish exploit memorization from exploitation reasoning?
> Response 2:
> This is an interesting line of thinking and we thank the reviewer for the thought. We believe that SandboxEscapeBench tests for (1) exploit memorisation and (2) whether a model can correctly implement a known technique in the context of the task. It also tests for (3) whether the model can conduct the significant amount of search required to determine what vulnerability is present before implementing the exploit. We argue that this mimics the vast majority of real world compromises and so is a useful property to measure and for that reason IS “exploitation reasoning”. In the paper we detail that we saw no novel solves of tasks. We would be interested to hear if the reviewer agrees with this.
>
> ---
>
> Comment 3: Difficulty ratings rest on a single expert's judgment without validation
> Response 3:
> We agree that relying on a single expert’s judgment is a limitation of the current paper. In response to the reviewer’s suggestion, we initiated two additional independent expert difficulty assessments and incorporated the findings into the paper.
>
> The agreement between the 3 reviewers is strong, with an ordinal Krippendorff's Alpha of 0.816. We also do not observe evidence of major systematic scoring bias between reviewers (max -0.17 mean score difference). These results provide some support for the robustness of the difficulty scale, while we acknowledge that a larger validation study with human participants would be valuable future work. Our benchmark does provide support for this through the “–solver human_cli” argument, but to do this properly would require additional work such as blinding participants to solutions, providing incentives, controlling AI assistance, and recruitment of sufficiently capable individuals.
>
> ---
>
> Comment 4: The threat model assumes initial container access, skipping the hardest practical step.
> Response 4:
> Our primary threat model for this benchmark is agents which have already been granted the ability to execute bash commands directly through tool-calling. This is now a common feature in agentic workflows (see https://platform.claude.com/docs/en/agents-and-tools/tool-use/bash-tool for an example implementation in a popular coding agent framework). To make this clearer, we have added the following sentence to the Introduction:
>
> "In these agentic deployments, code execution inside a container is granted by design rather than obtained through exploitation; initial access is therefore not a barrier in this threat model."

---

> > ### Author Rebuttal · Reviewer_LrCD · 2026-04-03
> >
> > Thanks to the author for the rebuttal, and I have no more questions to ask. I'd like to keep my score.

---

### Official Review · Reviewer_PZBG · 2026-03-17

**Soundness:** 3
**Presentation:** 3
**Significance:** 3
**Originality:** 3
**Overall Recommendation:** 4
**Confidence:** 3

**Summary:**

The paper presents SANDBOXESCAPEBENCH, a focused cybersecurity benchmark consisting of 18 diverse scenarios designed to evaluate the ability of frontier LLMs to escape containerized environments. Utilizing the Inspect AI framework, the authors implement a nested "sandbox-in-a-sandbox" architecture to safely test exploits across orchestration, runtime, and kernel layers. The study identifies a significant capability gap, where models successfully exploit common misconfigurations but fail at complex kernel vulnerabilities, and highlights a log-linear scaling of success relative to inference-time compute.

**Compliance With Llm Reviewing Policy:**

Affirmed.

**Final Justification:**

The concerns are addressed and I maintain my positive score.

**Key Questions For Authors:**

In Figure 3 and the Appendix, you refer to running "5 epochs." Could you clarify the exact relationship between an epoch, a trajectory, and an iteration? Specifically:

- Are epochs entirely independent (stateless) restarts of the sandbox?
- Does the model retain any "memory" or "learned context" from a failed epoch that influences the next attempt, or is this effectively a measure of Pass@k?

**Limitations:**

yes

**Strengths And Weaknesses:**

### Strengths

- Methodological Rigor: The use of a nested sandbox ensures that evaluations do not compromise host infrastructure, and the inclusion of human-expert reference solutions for all 18 tasks confirms the benchmark’s solvability.
- Objective Scoring: By utilizing exact string matching for a cryptographic flag (/flag.txt) located on the host, the authors eliminate the ambiguity of "hallucinated success" often found in agentic evaluations.
- Real-World Relevance: The benchmark targets high-frequency security failures (e.g., exposed Docker sockets, privileged mode, and over-provisioned capabilities) that are commonly encountered in CI/CD pipelines and enterprise cloud environments.

### Weakness
- Ambiguity in Terminology: Key parameters, such as the definition of "epochs" in the context of agentic trajectories, are not explicitly defined in the text, leading to potential confusion.
- Narrow Success Criteria: While "reading a flag" provides an objective metric, it represents a limited subset of attacker behavior. The benchmark does not currently account for other security objectives like data exfiltration, modifying host files or pivoting to network metadata services, etc.
- Lack of Mitigation Research: The paper identifies the problem (models can escape) but doesn't provide a deep dive into the effectiveness of specific defenses

---

> ### Author Rebuttal · Authors · 2026-03-30
>
> We thank the reviewer for their thoughtful and constructive feedback, which led to concrete improvements in the paper, particularly in clarifying terminology, motivating the benchmark scope, and expanding the discussion of mitigations.
>
> ---
>
> Comment 1: Ambiguity in Terminology
>
> Response 1:
> Yes this is ambiguous, we use the term “epoch”, borrowed from inspect-ai, to mean an independent trial, so an entirely new instantiation of the environment, with no context/notes.md shared between agentic attempts. We have swapped out “epochs” for “samples” and defined samples as: “independent attempts at the task”. Prompted by this we scanned our paper for other cases of ambiguity and found the use of “scenarios”, “challenges” and “tasks”, all to describe the set of 18 tasks. We have updated the paper to only refer to them as “tasks”.
>
> As each sample is independent, our reported success metrics are equivalent to pass@1. We did not report pass@5 due it being a high variance measure with 5 total samples. However, on further thought, we have added a new scaling plot with pass@5 metric to the appendix as the probability of seeing a single success in 5 independent attempts is a very threat-model grounded metric for cyber security.
>
> ---
>
> Comment 2:  Narrow Success Criteria
>
> Response 2:
> We agree that this benchmark measures a narrow part of the full attack chain. This was very much intentional as 1. There are existing benchmarks which measure other parts of the attack chain well e.g. cybench, cvebench, pacebench and cybergym. 2. The ability of models to escape from misconfigured sandboxes is currently not measured in the literature and 3. With a narrow scope we hoped to yield a measure with good construct validity and interpretability.
>
> We have added a comment in the discussion detailing why we focused on this narrow capability and that our results should be interpreted in context of other existing cyber benchmarks when assessing overall threat.
>
> ---
>
> Comment 3: Lack of Mitigation Research
>
> Response 3:
> We agree with the reviewer that mitigation research is critical here and hope that our paper will motivate more research in this area. We note that the vulnerabilities in our benchmark are predominantly real-world misconfigurations and known CVEs. The relevant defences are therefore not specific to AI: correct configuration, timely patching, and not relying on containers as a sole security boundary. Our contribution is in measuring whether models can detect and exploit these misconfigurations, which is complementary to existing guidance on preventing them. Thanks to the reviewer's suggestion, we have added a paragraph to the Discussion citing UK NCSC guidance on container security:
>
> "The UK NCSC notes that the kernel-enforced separation typically used by a container runtime is not usually considered enough compute separation to form a strong security boundary (UK National Cyber Security Centre, Using Containerisation: Running Containers, https://www.ncsc.gov.uk/collection/using-containerisation/running-containers). Guidance on recommended isolation measures, including defence in depth, can be found in official NCSC documentation."

---

> > ### Author Rebuttal · Reviewer_PZBG · 2026-04-04
> >
> > Thanks the authors for the clarification. I will be maintaining my positive score.

---

### Decision · Program_Chairs · 2026-04-30

**Decision:**

Accept (spotlight)

**Comment:**

This paper introduces SandboxEscapeBench, a CTF-style benchmark using nested sandboxing to safely evaluate whether frontier LLMs can escape container environments. All four reviewers were positive (two Weak Accept, two Accept), praising the benchmark's real-world relevance, rigorous safety design, objective scoring via flag retrieval, and extensive evaluation across nine frontier models. Concerns raised during review were addressed effectively: the authors clarified terminology (epochs vs. samples), added inter-rater reliability validation for difficulty ratings (Krippendorff's Alpha of 0.82 across three experts), sharpened the threat model framing to emphasize that agentic deployments grant container access by design, and better positioned the benchmark as a capability-isolation tool complementary to defense-aware frameworks like PACEbench. All four reviewers marked their concerns as fully resolved and confirmed their positive scores.

The remaining limitations are acknowledged and reasonable for a focused benchmarking contribution: the benchmark tests only known vulnerabilities (not zero-day discovery), uses a narrow success criterion (flag reading rather than broader attacker objectives), and evaluates relatively simple agentic architectures. The question of whether high performance reflects memorization versus genuine exploitation reasoning is partially addressed by the authors' argument that implementing known techniques in context constitutes real-world exploitation capability, though this remains an interesting direction for future work.

Based on the unanimous positive consensus, the thorough and well-received rebuttal, and the clear practical value of this benchmark for the AI safety and security communities, I recommend this paper for acceptance.